# Estimating Importance of Highly Correlated Features Using Matrix Decomposition

## Abstract

Hyperspectral images contain a large volume of source data that exhibits high correlations along neighboring spectral bands. This makes it necessary to select the most informative features among correlated groups of features to effectively solve various machine learning problems. A method of feature importance evaluation for hyperspectral image data is proposed. This method combines iterative training of decision tree classifiers based on spectral features with matrix decomposition to overcome sparsity. Decision trees provide intrinsic feature selection mechanism but only a small number of features are usually taken into account by the CART algorithm for training a single decision tree classifier instance. Furthermore when features are highly correlated (e.g., Pearson $\rho > 0.8$), tree-based methods like Random Forest or XGBoost arbitrarily assign importance to one feature while suppressing others, as they redundantly capture the same signal. The considered method is compared with several tree based methods for feature importance evaluation such as vanilla Gini impurity decrease and more complicated Boruta algorithm. The features are highlighted using a classification algorithm for thick cloud classification based on the marked-up satellite data. Classification accuracy testing based on significant features is performed for different types of surfaces for the set of several single images.

## 1 Introduction

Hyperspectral images (HSI) usually acquired from satellites capture light spectra in numerous narrow wavelength ranges. This approach boosts their informational content compared to conventional RGB images. In image processing tasks like cloud detection, monitoring, agriculture and environmental protection (Borzov & Uzilov, 2016), the large volume of HSI data and correlations across neighboring spectral bands require selection of the most informative features.

The existing methods of feature selection assess both feature properties and target variable relationships, covering forward selection and backward elimination methods, exhaustive search, and machine learning-based techniques (Guyon & Elisseeff, 2003). Feature importance evaluation can be done directly by analysing the pair relations between feature and target variable. This approach is generally called filter methods. Correlations between adjacent spectral bands gives folks not only to the idea of using dimentionality reduction algorithms for image compression but also to select relevant features (Myasnikov, 2017) from which PCA is the most popular (Zimichev et al., 2014). This approach along with direct analysis of the correlation matrix doesn't require additional labeling. Such methods as mutual information scores and ANOVA F-value are also very popular filter methods that takes into account labels but doesn't rely on training algorithms. Though these methods are computationally efficient they ignore feature interdependence which can lead to the selection of redundant features that are highly correlated with each other (Zhu et al., 2007). That is why, to select features more effectively, information from the results of HSI classification, in combination with the existing labeling of the source data, is needed. Machine learning (ML) models are widely used to evaluate the importance of features, especially in cases of feature correlations.

Model-specific methods for feature selection integrate the feature selection process directly into the model training phase. This approach offers good balance of performance and computational cost (Saito et al., 2018). There are wrapper and embedded model-specific methods. Wrapper methods identify the optimal subset of features by evaluating their different combinations using a specific

predictive model. Models such as the recursive feature elimination (RFE) select features iteratively, maximizing the performance of the classification model. Although effective in finding optimal subsets, these methods can be computationally demanding (Zubair et al., 2024). Embedded methods integrate feature selection directly into the model training process itself, using mechanisms like regularization penalties to perform selection simultaneously with parameter estimation. Examples of embedded methods include Lasso (Least Absolute Shrinkage and Selection Operator) and Ridge regression (Fira et al., 2025), neural networks with learnable drop layer (JimÉnez-Navarro et al., 2024), sparse principal component analysis (Seghouane et al., 2019), tree-based methods (Tuv et al., 2009).

Decision trees provide an intrinsic feature selection mechanism (Breiman et al., 1984), effectively handling non-linear data and correlated features Kohavi & John (1997), and offering inherent explainability Mishra (2022). Decision tree based models like Random Forests and Gradient Boosting provide feature importance scores based on how much a feature contributes to reducing impurity (e.g., Gini impurity decrease) or variance at each split in the decision trees. More complicated Boruta mathod determine feature relevance by comparing the importance of an original feature with the importance of permutated counterparts known as shadow features and functions as a wrapper algorithm built around the Random Forest classifier (Kursa & Rudnicki, 2010).

In general model specific methods give better results than filter methods but their significant drawback is that the selected set of features is inherently tied to the specific ML algorithm being used (Islam et al., 2022). Stochastic nature of some ML models can lead to reproducibility issues, with variations in feature importance rankings across different model training runs (Vos et al., 2024).

Model-agnostic methods offer greater flexibility and are widely used for interpreting complex "black-box" models. Model-agnostic feature selection methods can be applied to any machine learning model, as they are independent of the model's internal workings (Khan et al., 2025). These methods assess the relevance of features based on their intrinsic properties and their relationship with the target variable, without considering a specific predictive model. Examples of model-agnostic techniques include methods based on eXplainable Artificial Intelligence (XAI) like Permutation Feature Importance (PFI) (Flora et al., 2024) and SHapley Additive exPlanations (SHAP) (Lundberg & Lee, 2017). These techniques offer flexibility and can be used to compare the importance of features across different models but still affected by feature correlations (Liang et al., 2024) especially PFI (Salih, 2025).

The more sophisticated method of feature importance evaluation is used the more computing time it requires. The more features are taken into account, the more their importance is affected by multicollinearity. This paper is dedicated to the research of the possible approximation for feature importance evaluated from sparse matrix that contains minimal set of values obtained by selection of different feature subsets. We suggest the algorithm of feature importance recovery by means of matrix factorization.

## 2 PROBLEM STATEMENT AND METHOD OF SOLUTION

While high-dimensional HSI data can be rich in information, it also introduces specific obstacles that often reduces the effectiveness of ML models. A family of techniques designed to overcome the curse of dimentionality are commonly addressed through a set of methods known as dimensionality reduction. In this paper the dimentionality reduction is considered as feature selection problem.

The purpose of this work is to propose the algorithm for selecting a limited set of spectral bands and derived features that yields the best prediction quality of dense cloud classification. A dedicated set of labeled images of the HYPERION sensor with a spatial resolution of 30 m and a spectral resolution of 10 m in the spectral range of 400–2500 nm was used. After converting the raw radiance data into reflectance values and eliminating zero channels as well as those corresponding to strong light absorption in water vapour, the selected hyperspectral images (HSI) were organised into a training set. The features of this set comprised reflectance values from spectral channels 8–224, combined with derived characteristics and other indices calculated on a pixel-by-pixel basis. Fig. 1 shows the mean spectral reflectance distribution for cloud and non-cloud pixels, suggesting that a classification algorithm could be developed to differentiate between these pixels based on their

spectral characteristics. In this study, in addition to reflectance values, normalized indices such as NDVI (Huang et al., 2021), NDSI (Jin et al., 2022) and NDWI (Gao, 1996) were used.

We evaluate the importance of features by assess the cloud classification accuracy of a logistic regression model depending on the choice of spectral channels and derived features. The iterative training of ML models with different subset of features is used here to approximate target variable. After training of ML model the corresponding feature importance can be evaluated. By evaluating different sets of features used in the training process, we derive a sparse matrix representing the feature importances for each instance of the trained ML models. This approach yields the following observations:

1. Minimal subset of features is considered because of correlations between them;

2. Training ML models using as many features as possible is computationally expensive;

3. As a result of training ML models, the resulting importance matrix is sparse.

Logistic regression as linear model is strongly affected by features correlation. Decision trees are better but training with large feature subset is still computationally expensive. The resulting importance matrix is sparse, which gives rise to the problem of zero value reconstruction. Such reconstruction is needed to overcome possible feature importance underestimation for correlated features when using Gini impurity decrease as a metric of such evaluation. If we treat feature importance evaluation by approximating from a few values from the sparse matrix one can see that features with high importance ...

So we reformulated it by the following way. The feature importances are modeled as the following matrix decomposition:

$$\hat{T} = PQ^T, \tag{1}$$

where $\hat{T}$ ($n_u \times n_i$) is the predicted importances corresponding to trained ML model instance $u$ and feature $i$, $P$ ($n_u \times n_f$) and $Q$ ($n_i \times n_f$) are latent factors that capture hidden preferences for features and train instances, respectively. The challenge to the matrix factorization problem is to find $P$ and $Q^T$. Basically, such an algorithm is going to be used to find latent factors that represent intrinsic feature attributes in a lower dimension (the number $n_f$ of latent factors is choisen beforehand). A learning approach is therefore developed to converge the decomposition results close to the observed importances as much as possible, while ensuring all importance values remain nonnegative. Additionally we introduce the feature bias matrix $\bar{\mu}$ as a correction of (1):

$$\hat{T} = \mu + PQ^T. \tag{2}$$

The feature bias matrix $\bar{\mu}$ supposed to capture the tendency of features to have importance higher (or lower) than the average. So $\bar{\mu}$ measures a trained model tendency to systematically overestimate

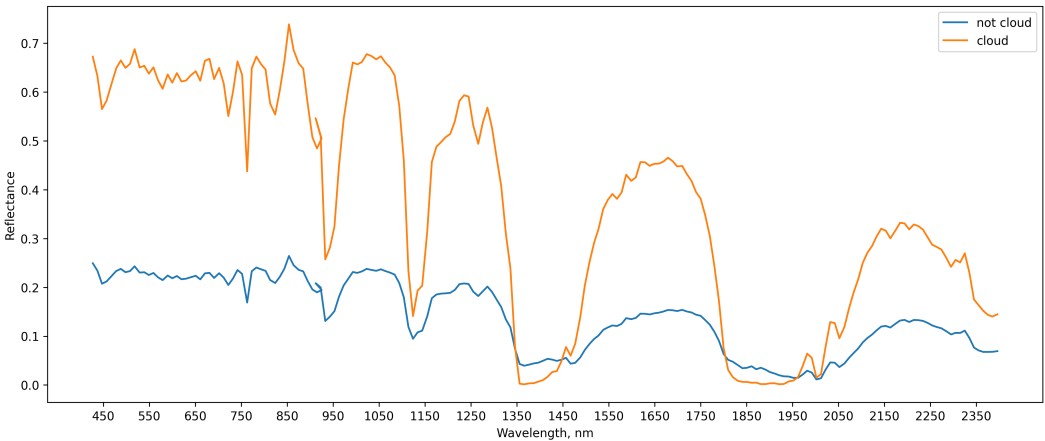

Figure 1: Mean spectral distribution of the reflectance corresponding to cloud and non-cloud pixels

or underestimate feature importances relative to the average across all trained ML model instances. Matrices $P$, $Q$ and $\bar{\mu}$ can be obtained through a regularized optimization procedure:

$$\sum_u \sum_i \left( (T_{u,i} - \hat{T}_{u,i})^2 + \lambda(\mu_{u,i}^2 + ||P_u||^2 + ||Q_i||^2) \right). \tag{3}$$

To bind classification accuracy with feature importance, the correlation between feature importances and classification error is added to the objective function:

$$\sum_u \sum_i \left( (T_{u,i} - \hat{T}_{u,i})^2 + R_{u,i} + \lambda(\mu_{u,i}^2 + ||P_u||^2 + ||Q_i||^2) \right), \tag{4}$$

where $\lambda$ is a regularization parameter, $e_u$ is the classification error of trained ML instance and

$$R_{u,i} = \sum_u \sum_i (e_u - \bar{e}_u)(\hat{T}_{u,i} - \mu_{u,i}) = \sum_u (e_u - \bar{e}_u) \sum_i p_u q_i^T. \tag{5}$$

Here $e_u - \bar{e}_u$ is the deviation of classification error $e_u$ from the average error of classification. The second term of equation (4) accounts for reducing the significance of features in the case of high classification error. If the classification error $e_u$ is below average the importance of features corresponding to the trained ML instance $u$ tends to increase; conversely, if $e_u$ is above average, the corresponding importance values tend to decrease.

Stochastic Gradient Descent used here to solve the problem (2) of matrix factorisation is an optimization algorithm in which the model parameters (in this case, the bias $\bar{\mu}$ and the factor vectors) are repeatedly updated by adding the negative of the gradients calculated with respect to the function (4) being optimized. The algorithm essentially performs the following steps for a given number of iterations:

$$\mu_{u,i} \leftarrow \mu_{u,i} + \gamma(\delta_{ui} - \lambda\mu_{u,i})$$
$$p_u \leftarrow p_u + \gamma\left((\delta_{ui} - 0.5 \cdot e_u) \cdot q_i - \lambda p_u\right)$$
$$q_i \leftarrow q_i + \gamma\left((\delta_{ui} - 0.5 \cdot e_u) \cdot p_u - \lambda q_i\right) \tag{6}$$

where $\gamma$ is the learning rate and $\delta_{ui} = r_{ui} - \hat{r}_{u,i} = r_{u,i} - (\mu_{u,i} + p_u q_i^T)$ is the error made by the model for the pair $(u, i)$.

## 3 RESULTS AND DISCUSSION

The input data for the matrix factorization algorithm is feature importance statistics evaluated by training Decision Tree classifier for different values of hyperparameters. For Decision Tree classifier the decrease of Gini impurity is usually used to assess the feature importance. By considering the accuracy associated with different feature sets, we can calculate the initial correlation between feature importance and training accuracy and sort features according to this value of correlation. Let's refer to this as correlation importance. The most important features include NDWI index and the limited set of features from the NIR and the lower SWIR wavelength range. There is little to no similarity between the feature importance rankings obtained via the Boruta algorithm and those from the present study, with the exception of the consistently high importance assigned to the NDWI feature. This can be explained by the use of shadow features in feature analysis via the Boruta algorithm not used here. However Fig. 2 shows the graph of the random forest model's accuracy versus the selected set of features. By choosing features with higher importance scores (starting from the top of the list in descending order of importance), we obtain models with higher training accuracy.

## 4 CONCLUSION

The significant number of channels in HSI, combined with feature multicollinearity, leads to challenges in selecting machine learning models, reducing their accuracy and interpretability. To address this issue for thick cloud classification, decision trees are employed with a selection of a limited number of significant features based on the iterative exclusion algorithm. The proposed method applied

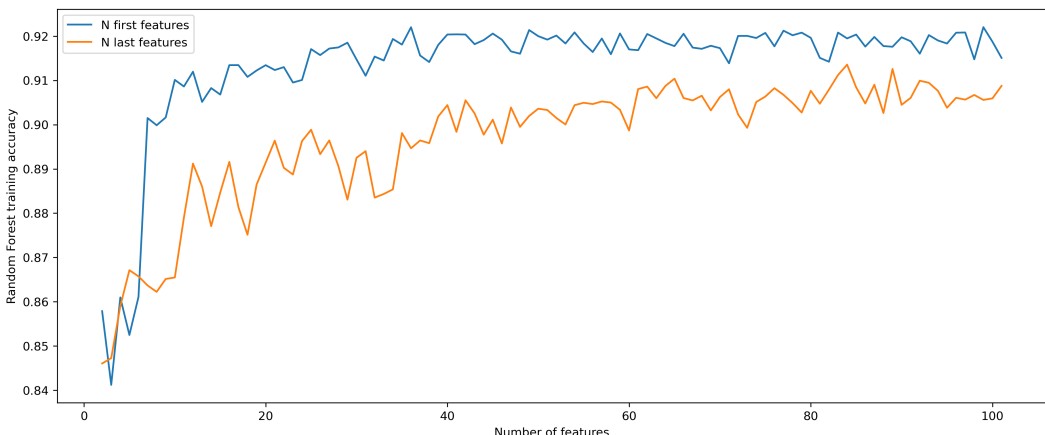

Figure 2: Random Forest training for the first and the last features by their correlation with training accuracy of different Decision Tree classifiers

to find the set of features that yields better prediction accuracy. The most relevant features include NDWI index, limited number of NIR bands and the lower part of SWIR spectrum range bands. Decision Tree model used to assess feature importance effectively handles correlated features avoiding redundancy and can be combined with forward feature selection to achieve more robust statistics. The final set of features selected after applying matrix factorization to overcome sparsity and taking into account the correlation between feature importance and model accuracy. This can be used to construct a classifier for recognizing thick clouds based on their spectral characteristics. Such classifier can be considered a baseline for more complex cloud classification models.

The ambiguity in feature importance rankings produced by traditional methods - such as permutation feature importance, SHAP values, tree-based methods, and linear model coefficients - underscores the need for a more empirically grounded approach. The proposed strategy of iterative feature elimination with feature approximation via matrix decomposition offers an alternative solution to this challenge. This approach not only reveals which features are truly critical for maintaining performance but also accounts for feature interactions and helps identify their optimal subset. One of the way to evaluate feature importance is to use Gini impurity decrease in combination with matrix factorization to overcome sparsity. The point of the present study is to use information about the classification accuracy to evaluate feature importance. This was done by the incorporation of the special correlation term in the objective function for optimization.

## ACKNOWLEDGMENTS

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
