# OpenReview forum: "ESTIMATING IMPORTANCE OF HIGHLY CORRELATED FEATURES USING MATRIX DECOMPOSITION"
_mathai.club/MathAI/2026/Conference — 2026 Oral_

### Official Review · Reviewer_jXL9 · 2026-03-12
**ESTIMATING IMPORTANCE OF HIGHLY CORRELATED FEATURES USING MATRIX DECOMPOSITION**

**Rating:** 7
**Confidence:** 5

**Review:**

Traditional algorithms (Random Forest, XGBoost) incorrectly distribute importance among highly correlated features, arbitrarily selecting one and suppressing the others. The authors propose a hybrid approach that combines iterative training of decision trees on feature subsets and matrix factorization to reconstruct a "sparse" importance matrix. The optimization objective function includes a special term that correlates feature importance with the classification error. If the error is below the mean, the importance of features in that run is increased.
The scientific novelty of this work is rated as moderately high due to its original approach to interpretability: 1. Matrix factorization for Feature Importance is used for the first time to modify importance values from sparse data obtained by iterating through feature subsets; 2. The integration of a model quality metric directly into the process of feature weight estimation via gradient descent.
Conclusion: The article proposes an elegant mathematical framework for overcoming the Gini Impurity limitation in tree-based models.

---

### Official Review · Reviewer_DFys · 2026-03-12
**Abstract “ESTIMATING IMPORTANCE OF HIGHLY CORRELATED FEATURES USING MATRIX DECOMPOSITION” submitted to MathAI 2026 Conference describes the algorithm for feature importance approximation taking into account the ML model accuracy.**

**Rating:** 6
**Confidence:** 3

**Review:**

Abstract “ESTIMATING IMPORTANCE OF HIGHLY CORRELATED FEATURES USING MATRIX DECOMPOSITION” submitted to MathAI 2026 Conference describes the algorithm for feature importance approximation taking into account the ML model accuracy. The Abstract is well -written and have some interest  for readers. Classification accuracy testing based on significant features is performed for different types of surfaces for the set of several single images. Routine hyperspectral images contain a large volume of data with  high correlations along neighboring spectral bands. The most informative features among correlated groups of features can effectively solve various machine learning problems based on analysis of much smaller number of data.
A new method of feature importance evaluation for hyperspectral image data is proposed in this work. This method consist of iterative training of decision tree classifiers based on spectral features with matrix decomposition to overcome sparsity. When features are highly correlated (e.g., Pearson ρ > 0.8), tree-based methods like Random Forest or XGBoost arbitrarily assign importance to one feature while suppressing others.  The method was compared with several tree based methods for feature importance evaluation such as vanilla Gini impurity decrease and more complicated Boruta algorithm.

---

### Official Review · Reviewer_CuXD · 2026-03-12
**Review of the paper "ESTIMATING IMPORTANCE OF HIGHLY CORRELATED FEATURES USING MATRIX DECOMPOSITION"**

**Rating:** 7
**Confidence:** 2

**Review:**

The paper makes a valuable contribution to the problem of feature importance estimation in the presence of high correlation, a particularly relevant issue for hyperspectral data.

The authors effectively demonstrate the limitations of standard tree-based methods (including Random Forest and XGBoost) in such scenarios and propose an original hybrid approach: iterative training on feature subsets, matrix factorization to recover distributed importance, incorporation of classification error feedback into the optimization process.

The work appears mathematically sound, and the claimed scientific novelty is convincing: the combination of matrix decomposition specifically with an iterative procedure for obtaining sparse importance estimates, together with the direct coupling of feature weights to model quality via a dedicated optimization term.

---

### Decision · Program_Chairs · 2026-03-14

**Decision:**

Accept (Oral)

**Comment:**

Dear Author(s),

On behalf of the Program Committee of the International Conference on Mathematics of Artificial Intelligence (MathAI 2026), we are pleased to inform you that your paper has been accepted for an oral presentation at MathAI 2026.

Your paper was evaluated through a rigorous two-stage review process involving both automated screening and expert review by members of the Program Committee. The reviewers recognized the quality and contribution of your work.

Presentation details:

- Format: Oral presentation (15–20 minutes + 5 minutes Q&A)
- Mode: You may present either in person (offline) at the conference venue in Sirius, Russia, or remotely via Zoom. Please indicate your preferred mode when confirming your participation.
- Conference dates: Marh 30 - April 3, 2026
- Website: https://mathai.club

Next steps:

1. Please confirm your participation and presentation mode by replying to this email mathai.club@yandex.ru no later than March 15, 2026 18:00 Moscow time.
2. If you plan to attend in person, the organizing committee will provide accommodation details separately.
3. Please prepare your final camera-ready manuscript according to the formatting guidelines available at https://mathai.club and upload it to OpenReview by March 15, 2026 18:00 Moscow time.

Should you have any questions regarding the program, logistics, or your presentation slot, please do not hesitate to contact us.

We look forward to your contribution to MathAI 2026.

With kind regards,

MathAI 2026 Program Committee
International Conference on Mathematics of Artificial Intelligence
https://mathai.club
OpenReview: https://openreview.net/group?id=mathai.club/MathAI/2026/Conference
Telegram: https://t.me/MathAI_club
Email: mathai.club@yandex.ru